# Sevoflurane, Propofol and Carvedilol Block Myocardial Protection by Limb Remote Ischemic Preconditioning

**DOI:** 10.3390/ijms20020269

**Published:** 2019-01-11

**Authors:** Youn Joung Cho, Karam Nam, Tae Kyong Kim, Seong Woo Choi, Sung Joon Kim, Derek J Hausenloy, Yunseok Jeon

**Affiliations:** 1Department of Anesthesiology and Pain Medicine, Seoul National University Hospital, Seoul National University College of Medicine, Seoul 03080, Korea; mingming7@gmail.com (Y.J.C.); karamnam@gmail.com (K.N.); ktkktk@gmail.com (T.K.K.); 2Department of Anesthesiology and Pain Medicine, SMG-SNU Boramae Medical Center, Seoul 07061, Korea; 3Department of Physiology, Department of Biomedical Sciences, Seoul National University College of Medicine, Seoul 03080, Korea; djmaya@snu.ac.kr (S.W.C.); physiolksj@gmail.com (S.J.K.); 4Department of Stem Cell Biology, School of Medicine, Konkuk University, Seoul 05029, Korea; 5Cardiovascular and Metabolic Disorders Program, Duke-National University of Singapore Medical School, Singapore 169857, Singapore; d.hausenloy@ucl.ac.uk; 6Hatter Cardiovascular Institute, Institute of Cardiovascular Science, University College of London, London WC1E 6HX, UK; 7Tecnologico de Monterrey, Centro de Biotecnologica-FEMSA, Nuevo Leon 64849, Mexico; 8Yong Loo Lin School of Medicine, National University of Singapore, Singapore 119228, Singapore; 9The National Institute of Health Research, University College London Hospitals, Biomedical Research Centre, London W1T 7DN, UK; 10National Heart Research Institute Singapore, National Heart Centre, Singapore 169609, Singapore

**Keywords:** ischemic preconditioning, remote ischemic conditioning, cardiac surgery, cardioprotection, ischemia-reperfusion injury, carvedilol, propofol, sevoflurane

## Abstract

The effects of remote ischemic preconditioning (RIPC) in cardiac surgery have been inconsistent. We investigated whether anesthesia or beta-blockers interfere with RIPC cardioprotection. Fifty patients undergoing cardiac surgery were randomized to receive limb RIPC (four cycles of 5-min of upper arm cuff inflation/deflation) in the awake state (no-anesthesia; *n* = 17), or under sevoflurane (*n* = 17) or propofol (*n* = 16) anesthesia. In a separate crossover study, 11 healthy volunteers received either carvedilol or no medication prior to RIPC. Plasma dialysates were obtained and perfused through an isolated male Sprague–Dawley rat heart subjected to 30-min ischemia/60-min reperfusion, following which myocardial infarct (MI) size was determined. In the cardiac surgery study, pre-RIPC MI sizes were similar among the groups (39.7 ± 4.5% no-anesthesia, 38.9 ± 5.3% sevoflurane, and 38.6 ± 3.6% propofol). However, post-RIPC MI size was reduced in the no-anesthesia group (27.5 ± 8.0%; *p* < 0.001), but not in the anesthesia groups (35.7 ± 6.9% sevoflurane and 35.8 ± 5.8% propofol). In the healthy volunteer study, there was a reduction in MI size with RIPC in the no-carvedilol group (41.7 ± 4.3% to 30.6 ± 8.5%; *p* < 0.0001), but not in the carvedilol group (41.0 ± 4.0% to 39.6 ± 5.6%; *p* = 0.452). We found that the cardioprotective effects of limb RIPC were abolished under propofol or sevoflurane anesthesia and in the presence of carvedilol therapy.

## 1. Introduction

New treatment strategies are required to improve clinical outcomes in patients with ischemic heart disease, the leading cause of death and disability worldwide. Remote ischemic preconditioning (RIPC), in which brief cycles of non-lethal ischemia and reperfusion are applied to an organ or tissue away from the heart, has been shown to protect the myocardium against lethal acute ischemia/reperfusion (IR) injury [1]. This strategy, which can be non-invasively applied by simply inflating and deflating a pneumatic cuff placed on the upper arm to induce cycles of brief IR (termed limb RIPC), has been shown to reduce procedure-related myocardial injury in patients undergoing elective percutaneous coronary intervention [2] as well as myocardial infarct (MI) size in ST-segment elevation myocardial infarction patients [3,4,5].

In the setting of coronary artery bypass graft (CABG) surgery, several studies have shown a reduction in peri-operative myocardial injury with limb RIPC as evidenced by attenuated release of cardiac biomarkers such as cardiac troponins [6,7]. However, three large clinical studies in cardiac surgery have failed to show any benefits of limb RIPC on post-surgical clinical outcomes [8,9,10]. The reasons for this are unclear but have been attributed to several factors including the use of propofol [11,12]. However, there has been no direct demonstration that propofol blocks the effects of RIPC [13]. It has also been suggested that beta-blockers (BBs) may interfere with RIPC by attenuating organ protection or altering RIPC-related signaling pathways [14].

The mechanisms underlying the cardioprotective effects of limb RIPC are not clear, although a neuro-humoral pathway linking the limb to the heart has been implicated [15]. In previous studies focused on the parasympathetic system, blocking these signaling pathways attenuated the effects of RIPC [15,16]. However, few studies have been conducted on the sympathetic nervous system. With respect to the humoral pathway, the cardioprotective effects of RIPC can be transferred between species [17,18,19,20]. Therefore, the effects of RIPC can be evaluated by transferring the protective factors from human plasma dialysate to the rat heart perfusion system subjected to acute IR injury [18].

We hypothesized that propofol attenuates the cardioprotective effects of RIPC in cardiac surgery patients. To evaluate our hypothesis, we used a Langendorff-perfused rat heart model to investigate whether propofol or volatile anesthetic, sevoflurane affects the cardioprotective effects of limb RIPC in cardiac surgery patients. In addition, we investigated the effects of BB, carvedilol treatment on RIPC-induced myocardial protection in healthy volunteers.

## 2. Results

Cardiac surgery patients were enrolled between 22 February 2017 and 16 January 2018. Among the 162 patients screened, 111 were excluded and 51 were allocated to the no-anesthetic, sevoflurane, and propofol groups (Figure 1a). One patient in the propofol group was excluded as he did not meet the inclusion criteria, thus 50 patients were included in the final analyses. In one patient in the sevoflurane group, there was an intraoperative event due to suspected injury of the left coronary artery. Therefore, data on troponin levels for this patient were excluded from the analyses. 

Eleven healthy volunteers were recruited between 17 May 2017 and 25 October 2017 and completed the protocol (Figure 1b). No complications related to RIPC or the use of carvedilol were detected. There was one experimental failure with the Langendorff model each in the no-anesthesia and carvedilol group, due to the sustained poor ventricular contractile function and the poor contractile function combined with sustained ventricular fibrillation, respectively. Therefore, these data were excluded from the analyses. Study participants’ characteristics are shown in Table 1 and Table 2.

According to our preliminary data, MI size of rat hearts (control group, *n* = 3) that were perfused for 30 min and then underwent IR injury, consisting of 30-min global ischemia followed by 60-min reperfusion, without treatment of human dialysate, was 41.1 ± 6.8%. For the control group, no other interfering substances but only neutral buffer solution was used to perfuse the isolated rat heart. In addition, we confirmed that the freezing process before cutting and staining the heart slices did not induce negative staining in sham hearts (*n* = 2) by performing negative control experiments, in which the hearts were excised, frozen at −20 °C for 30 min, cut into slices and then stained (Figure 2).

In the cardiac surgery study, pre-RIPC MI sizes were similar among the three cardiac surgery groups (39.7 ± 4.5% no-anesthesia, 38.9 ± 5.3% sevoflurane, and 38.6 ± 3.6% propofol; *p* = 0.785). However, MI size was significantly reduced after RIPC in the no-anesthesia group (post-RIPC MI size, 27.5 ± 8.0%; mean difference −12.1%; 95% CI −17.4%, −6.9%; *p* < 0.001), but not in the two anesthetic groups (35.7 ± 6.9% sevoflurane and 35.8 ± 5.8% propofol) (Figure 3). Post-RIPC MI size in the sevoflurane group was inferior to that of the no-anesthesia group (mean difference, −8.1% (one-sided 97% CI −13.5%, ∞), *p* > 0.999; non-inferiority test), but did not differ from that of the propofol group (*p* = 0.957).

In the healthy volunteer study, there was a significant reduction in MI size with RIPC in the no-carvedilol group (41.7 ± 4.3% pre-RIPC vs. 30.6 ± 8.5% post-RIPC; mean difference −10.6%; 95% CI −14.7%, −6.5%; *p* < 0.0001) (Figure 4). However, in the carvedilol group, there was no significant difference in MI size (41.0 ± 4.0% pre-RIPC vs. 39.6 ± 5.6% post-RIPC; mean difference −1.6%; 95% CI −5.9%, 2.8%; *p* = 0.452). There was no carry-over effect of the first period treatment (*p* = 0.216).

There were no differences in postoperative variables among the groups of patients (Table 3).

## 3. Discussion

In this study, when patients were not under anesthesia, the dialysate from the patients post-RIPC reduced MI size of rat hearts following acute IR injury. However, RIPC-induced cardioprotection was attenuated when the patients were under either sevoflurane or propofol anesthesia. There were no differences between the two anesthetics. In healthy volunteers, the use of carvedilol attenuated the RIPC-induced cardioprotection.

Since it was first described in 1993 [21], many trials have applied RIPC in various settings, showing conflicting results. RIPC applied to human subjects has elicited myocardial protection and improved clinical outcomes in patients undergoing primary coronary intervention [2,22,23]. However, in patients undergoing cardiac surgery involving CABG and/or cardiac valve surgery, RIPC has not provided beneficial implication on clinical outcomes beyond the reduction in myocardial injury markers [24,25]. In three large multi-center trials involving more than 1000 patients undergoing cardiac surgery, RIPC failed to have benefits on composite clinical outcomes [8,9,10]. The reason for the discrepancy in clinical results has not yet been well investigated and remains undetermined [19,25].

There have been many attempts to identify possible confounding factors that may interfere with RIPC. In the aforementioned trials, which all showed a neutral effect of RIPC on outcomes after cardiac surgery, most patients received propofol during RIPC [8,9,10], and the use of propofol was proposed as the main factor responsible for the results. Propofol (2,6-diisopropylphenol) is a lipid-soluble anesthetic agent that contains a phenolic structure similar to the phenol-based free radical scavenger α-tocopherol, which is a natural antioxidant [26]. In an in vivo rat study, propofol attenuated small intestinal IR injury by suppressing oxidative stress [27]. Because treatment with free radical scavengers abolished the volatile-induced preconditioning effects [28,29], it is plausible that the free radical scavenging ability of propofol might attenuate the effects of RIPC [30].

There have been several clinical trials supporting this hypothesis. In a previous small trial in patients undergoing CABG, RIPC reduced postoperative cardiac troponin I (cTnI) under isoflurane anesthesia, but did not under propofol anesthesia [31]. In another trial, sub-group analyses showed that the use of propofol abrogated the effects of RIPC after CABG [7]. Furthermore, when the use of propofol was precluded, RIPC reduced acute kidney injury in high-risk surgical patients [32].

In this study, we found that RIPC did not protect myocardium against IR injury in isolated rat heart under either sevoflurane or propofol anesthesia. According to our results, propofol did not act differently from sevoflurane in attenuating the effects of RIPC. There are several possible explanations. First, the concentration of propofol administered during the study period (approximately 40 min) may not have reached the concentration required to have cardioprotective effects [33]. Second, both sevoflurane and propofol attenuated oxidative stress by reducing reactive oxygen species [34], suggesting that both might interfere with RIPC. Third, general anesthesia itself, regardless of anesthetic agents, may suppress the neurogenic pathways of RIPC. Traditionally, inhibiting the afferent and efferent nerve pathways are two of the four components of general anesthesia [35]. Thus, it is possible that the neurogenic pathways of RIPC could be suppressed during general anesthesia. In a recent Cochrane review, subgroup analyses provided no definite conclusion on whether propofol interfered with or inhibited the effects of RIPC [36].

The results of our study are not consistent with a recent work by Behmenburg and colleagues [37], who investigated the impact of different anesthetic regimens on RIPC in an animal model. In that study, pentobarbital and sevoflurane applied to rats preserved the effects of RIPC, whereas propofol completely abolished the protection when comparing MI sizes. However, while animal experiments and pre-clinical studies have promising results, the clinical application of RIPC is more complex. Human patients are in a very different condition from healthy experimental animals. Most clinical studies, including ours, are conducted in diseased patients undergoing surgical procedures. Our patients are typically elderly, have various comorbidities and have been taking comedications. RIPC dialysate from diabetic patients with peripheral neuropathy mitigated cardioprotective effects [18], and comedications such as angiotensin receptor blocker could interfere with ischemic conditioning [38]. Accordingly, under sevoflurane anesthesia, RIPC elicited protective effects in children with congenital heart defects [39], but not in adults with comorbidities [40]. Nevertheless, the plasma dialysate from the patients received RIPC without anesthetics showed significant cardioprotection in rat hearts in our study.

In this study, a single administration of carvedilol abrogated the effects of ischemic conditioning in healthy volunteers. Among several explanations for this observation, BB may have blocked the neurogenic pathways of RIPC, as muscarinic receptor blockers or vagotomy blocked the effects of RIPC [15,16]. Similar to propofol, carvedilol acts as a free radical scavenger [41,42], thereby inhibiting the effects of ischemic conditioning in animal studies. Thus, there is also a possibility that carvedilol might have blocked effects of RIPC by its anti-oxidative properties regardless of blockade of beta-adrenoceptors in this study. Chronic pre-treatment with propranolol and nipradilol abrogated the effects of ischemic preconditioning in rat hearts [43]. A brief treatment of short-acting BB, esmolol, also attenuated the effects of ischemic preconditioning in rabbit hearts [14]. In addition, beta-adrenergic blockade with atenolol abrogated the beneficial effects of preconditioning in isolated mouse hearts [44]. However, retrospective analyses of clinical studies showed no impact of BB on RIPC-induced cardioprotection in patients undergoing CABG surgery [45]. Therefore, further studies are required to elucidate the effects of carvedilol on RIPC.

In addition, opioid receptors are implicated in myocardial conditioning and cardioprotective responses [46]. Continuous administration of remifentanil provided cardioprotection in in vitro human myocardium against hypoxia reoxygenation stress [47]. On the other hand, sustained administration of remifentanil inhibited preconditioning-induced infarct limitation following one cycle of preconditioning (5-min ischemia and 5-min reperfusion), but did not affect MI size reduction elicited by two cycles of preconditioning in rabbit hearts [48]. In this study, we investigated the effects of sevoflurane and propofol on RIPC-induced cardioprotection in a pragmatic manner that is commonly used in clinical anesthetic practice for cardiac surgery patients. Nevertheless, we cannot completely exclude any influence of remifentanil administered during RIPC in this study, and further studies are required to determine the effects of administered remifentanil during RIPC in cardiac surgery patients.

This study has several limitations. First, this study does not provide insight on mechanism by which propofol, sevoflurane, or carvedilol treatment diminishes the production, transfer or translation of RIPC signals. However, these issues are beyond of the scope of this study and should be investigated in further studies. Second, the sample size calculation was based on the MI size of the rat hearts treated with plasma dialysate. Accordingly, the limited number of enrolled patients elicited heterogeneity in the baseline characteristics. Future studies may limit the study population to a homogenous group to reduce the heterogeneity but this may also reduce external validity. Third, the included patients mostly had valvular heart diseases, whereas most previous studies on RIPC involved patients with ischemic heart disease [9,10,22,23]. However, patients undergoing valve surgery are also susceptible to IR injury and are candidates for organ protection, although their disease entity is diverse. Fourth, calculation of the MI size of rat hearts did not take into account the weight of the heart slices, and only areas were used. Lastly, clinical parameters apart from the primary outcome were not adequately powered. Moreover, the anesthetic technique was standardized with propofol-based anesthesia in all groups after RIPC. Therefore, we should be careful in interpreting the results on other outcomes, such as troponin levels.

## 4. Materials and Methods 

### 4.1. Ethical Approval

The studies in patients and healthy volunteers were approved by the Institutional Review Board of Seoul National University Hospital (1605-079-761, 20 July 2016 and 1702-031-829, 3 March 2017) and registered at clinicaltrials.gov (NCT02932722 and NCT03169426) before enrollment. The study was performed according to Good Clinical Practice guidelines and the principles of the Declaration of Helsinki. All participants provided written informed consent and were allowed to withdraw their consent at any time. All animal experimental protocols were approved by the Institutional Animal Care and Use Committee (IACUC) of Seoul National University (SNU-160812-2-2, 20 April 2017 and SNU-170417-22, 4 May 2017) and were in accordance with the Guide for the Care and Use of Laboratory Animals published by the US National Institutes of Health (NIH, Bethesda, MD, USA).

### 4.2. Study Population

For the cardiac surgery study, patients aged 20–80 years old and undergoing cardiac surgery using cardiopulmonary bypass at Seoul National University Hospital were eligible for inclusion. We did not include over-octogenarian patients, as any possible changes in neuronal or humoral responses to RIPC in these patients were not clearly identified. Patients were excluded if they met any of the following criteria: undergoing descending thoracic aorta surgery; engaged in strenuous exercise, alcohol or caffeine intake within 24 h prior to surgery; preoperative left ventricular (LV) ejection fraction <30%; uncontrolled hypertension or diabetes mellitus; severe renal or hepatic dysfunction; receiving hemodialysis; presence of arterio-venous fistula on arms; peripheral vascular or nerve disease; coagulopathy; preoperative use of BBs; preoperative use of mechanical circulatory support devices; emergent or redo operation; refusal to enroll; or were pregnant. 

For the healthy volunteers study, healthy male volunteer aged 20–45 years old were recruited by local advertising. Subjects were excluded based on the following criteria: use of any prescribed or over-the-counter drugs; allergic history to any food or medication; body mass index < 18 or > 30 kg/m^2^; baseline systolic blood pressure (BP) > 150 mmHg or < 100 mmHg, or diastolic BP > 100 mmHg or < 50 mmHg; strenuous exercise, consumption of tobacco or alcohol or caffeine-containing materials within 24 h prior to study; use of herbal medications within 14 days; presence of problems such as vascular abnormalities on the upper extremities that precluded RIPC; or did not consent to participate.

### 4.3. Randomization and Masking

Enrolled cardiac surgery patients were randomized to one of the following three treatment groups: no-anesthesia, sevoflurane, or propofol on the day of surgery (Figure 1a). Block randomization (blocks of six or nine) was conducted using a computer-generated randomization program by an independent researcher to allocate patients in a 1:1:1 ratio. Group assignment was concealed in opaque envelopes and blinded to surgeons and investigators involved in animal experiments and data analyses. The animal studies and data analyses were undertaken by investigators blinded to the treatment allocation.

Using a randomized, two-treatment, two-period crossover study design, healthy male volunteers received either a single oral dose of carvedilol 12.5 mg (carvedilol group) or no medication (no-carvedilol group). Randomization codes were generated by a computer, sealed in opaque envelops and assigned to the subjects after enrollment. Each study protocol was separated by a washout period of six days (Figure 1b). 

### 4.4. Cardiac Surgery Study Design

Without premedication, patients were monitored with 5-lead electrocardiogram, non-invasive BP, pulse oximetry, bispectral index, and cerebral oximetry. Arterial cannulation for continuous BP monitoring was established at the radial artery. In the no-anesthesia group, RIPC was performed in the awake state before anesthesia on the day of surgery (Figure 5). General anesthesia was provided to the patients in the no-anesthesia group after RIPC study protocol during the surgery. In the sevoflurane and propofol groups, RIPC was commenced after the induction of anesthesia (Figure 5). Anesthesia was induced with midazolam 1 mg/kg in the sevoflurane group, while effect-site concentration of propofol (Fresofol 2 MCT 2%, Fresenius Kabi, Graz, Austria) 4 μg/kg/min with target-controlled infusion in the propofol group. During RIPC, maintenance doses of anesthetics were 2 vol% of sevoflurane (Sojourn, Piramal Critical Care Inc., Bethlehem, PA, USA) and 4 μg/kg/min of propofol in the sevoflurane and propofol groups, respectively. Continuous infusion of remifentanil (Ultiva, GlaxoSmithKline, Middlesex, UK) was used in both groups. After completion of RIPC, anesthesia was maintained with propofol and remifentanil in all groups. The RIPC protocol consisted of four cycles of 5-min ischemia and 5-min reperfusion induced by inflating the pneumatic cuff on the upper arm to 200 mmHg, and deflating the cuff, respectively.

### 4.5. Healthy Volunteer Study Design

Study protocols were performed in a quiet room in the supine position. Subjects were asked to remain nil per os for more than 4 h prior to each investigation, and were allowed to rest for 5 min in the supine position prior to the investigation. During the study, three-lead electrocardiogram, non-invasive BP, and pulse oximetry were monitored. RIPC was commenced 90 min after taking carvedilol in the carvedilol group, or after 5 min of rest in the no-carvedilol group (Figure 6).

### 4.6. Preparation of Human Plasma Dialysate

Blood samples (30 mL) were obtained from the participants before (pre-RIPC) and after completion of RIPC (post-RIPC). In the cardiac surgery patients randomized to receive sevoflurane or propofol, pre-RIPC samples were collected before anesthetic administration. The blood was collected in sodium heparin tubes and immediately centrifuged at 3000 rpm for 20 min at room temperature (Figure 7). The plasma fraction was carefully separated without disturbing the buffy coat, and was dialyzed across a 12–14 kDa dialysis tubing membrane (Spectra/Por, Spectrum Laboratories, Inc., Rancho Dominguez, CA, USA) against a 20-fold volume of modified Krebs–Henseleit buffer (KHB) for 24 h at 4 °C. If the plasma could not be used immediately, it was stored in a −80 °C freezer. Modified KHB for dialysis consisted of 118 mM NaCl, 4.7 mM KCl, 1.1 mM MgSO_4_·7H_2_O, 1.2 mM KH_2_PO_4_, and 1.8 mM CaCl_2_·2H_2_O. Prior to perfusion of the rat heart, the dialysate was supplemented with 25 mM NaHCO_3_ and 11 mM d-glucose, gassed with a 95% O_2_ and 5% CO_2_ mixture, and equilibrated to 37 °C. Finally, the pH was adjusted to 7.4.

### 4.7. Langendorff-Perfused Rat Heart Model

Male Sprague–Dawley rats, aged 9–11 weeks and weighing 250–350 g, were housed under specific pathogen-free conditions on a 12-h/12-h light/dark cycle with free access to food and water. The environmental temperature and humidity were maintained at 24–25 °C and 40–60%. All animals were cared for in strict compliance to the Guidelines for Care and Use of Laboratory Animals issued by the IACUC of Seoul National University. Rats were anesthetized with intraperitoneal injections of 60 mg/kg of pentobarbital sodium or inhalation of 6–8 vol% sevoflurane. Among the 100 anesthetized rats used in the cardiac surgery study, 10 were given pentobarbital sodium and the remaining rats were given sevoflurane due to the shortage of pentobarbital in our animal laboratory. Forty-four rats used in the healthy volunteer study were anesthetized with sevoflurane. The duration of the inhalation of sevoflurane until the opening of the thorax was less than 2 min. Therefore, although we cannot exclude the possibility of any effects of inhaled sevoflurane in rats completely, any additive cardioprotective effects induced by this brief period of inhalation would have affected MI size equally among the treatment groups. Heparin (100 IU/kg) was administered via a lateral tail vein. After confirming the loss of pedal reflex, the hearts were rapidly excised via a clamshell thoracotomy, mounted on a Langendorff-apparatus within 1 min via ascending aorta cannulation, and perfused with KHB solution in a retrograde, non-recirculating manner. KHB consisted of 118 mM NaCl, 25 mM NaHCO_3_, 11 mM d-glucose, 4.7 mM KCl, 1.1 mM MgSO_4_·7H_2_O, 1.2 mM KH_2_PO_4_, and 1.8 mM CaCl_2_·2H_2_O, was gassed with a 95% O_2_ and 5% CO_2_ mixture, and adjusted to a pH of 7.4 at 37 °C. Following 10-min stabilization, the heart was perfused with pre- or post-RIPC dialysate for 15 min and then washed out for 5 min prior to 30-min no-flow global ischemia and subsequent 60-min reperfusion (Figure 8) [49,50,51]. Each heart was treated only with one certain plasma dialysate from one human subject. The temperature of the heart was maintained at 37 °C. Hearts were excluded if they met one of the following exclusion criteria: time to perfusion >3 min; unstable contractile function; significant arrhythmic duration >3 min; or heart rate <100 or >400 beats per minute.

At the completion of reperfusion, the hearts were removed from the apparatus and placed at −20 °C for 30 min [50,51,52]. The semi-frozen hearts were cut into 5–6 slices transversely, each 1–2 mm thick. Then the slices were incubated in 1% 2,3,5-triphenyltetrazolium chloride in 0.1 M sodium phosphate buffer adjusted to pH 7.4, at 37 °C for 15 min to distinguish viable tissue (brick red color) and infarcted tissue (pale white color). The slices were fixed in 10% formalin solution for 24 h for contrast enhancement. Then, both sides of the slices were digitally scanned for planimetric analyses by a blinded observer using ImageJ software (ver 1.51, NIH, Bethesda, MD, USA). Because the hearts were subjected to global ischemia, the total cross-sectional LV areas were defined as the total areas at risk, and MI size was expressed as a percentage of the total area of the LV [20,51,52,53]. The reported MI size is the mean of all MI measurements from both sides of all individual slices.

### 4.8. Study Endpoints and Sample Size Calculation

The primary endpoint of the cardiac study was MI size of the rat heart perfused with human dialysate. Previously, the MI size in a Langendorff IR injury model using human dialysate was 27.4 ± 3.8% [54]. Based on this, we hypothesized that the MI size in the sevoflurane group would be comparable to that of the no-anesthesia group, and that the MI of the propofol group would differ from that of the sevoflurane group. We calculated the sample size at a significance level of 2.5% for each hypothesis with a power of 80%. For testing non-inferiority of sevoflurane compared to the no-anesthesia regarding post-RIPC MI size, 13 patients per group were required for non-inferiority margin of 5%, which was chosen by clinical relevance, including a 10% dropout rate. For comparison of the sevoflurane and propofol groups, we calculated that 17 patients were required per group to detect a clinically relevant percent change in MI of 20% between groups, assuming a 10% dropout rate. Based on these calculations, we decided to enroll 17 patients per group (a total of 51 patients in the study).

Secondary endpoints included the difference in MI changes between the carvedilol and no-carvedilol groups. Other study endpoints in cardiac surgery patients included postoperative variables, such as peak level of cTnI during postoperative 72 h, amount of chest tube drain during the first postoperative 24 h, newly occurred atrial fibrillation, duration of intensive care unit stay, and postoperative hospital stay. Evaluation of cTnI was performed using the Abbott ARCHITECT analyzer (Abbott Laboratories, Abbott Park, IL, USA), which has a limit of detection of 0.001 ng/mL and an overall 99th percentile of 0.026 ng/mL. A cTnI level higher than 0.028 ng/mL was considered abnormal. New instances of postoperative atrial fibrillation were identified based on the presence of sustained atrial fibrillation on patients’ electrocardiographic monitors that required medical treatment or direct-current cardioversion.

### 4.9. Statistical Analyses

Data are expressed as mean ± SD, median (IQR (range)), number (proportion), or mean difference with 95% confidence intervals (CIs). The normality of the data was tested using the Kolmogorov–Smirnov test and Shapiro–Wilk test. Continuous variables were compared using the independent *t*-test, one-way analysis of variance, or Mann–Whitney U test. Categorical variables were compared using Pearson’s chi square test or Fisher’s exact test. For the analysis of primary endpoint, non-inferiority of sevoflurane to no-anesthetic with regard to post-RIPC MI size was evaluated using one-sided *t*-test at a significance level of 0.025, and post-RIPC MI size between sevoflurane and propofol group was compared using two-sided *t*-test at a significance level of 0.025. In the non-inferiority test of sevoflurane comparing to no-anesthetic, a *p*-value to reject the null hypothesis of mean post-RIPC MI size in the no-anesthetic group is at least 5% less than those in the sevoflurane group was reported. For comparison of pre- and post-RIPC MI size within a group, paired *t*-test with correction for multiple testing was used. For the secondary endpoints, comparison of MI size changes between the carvedilol and no-carvedilol group in the 2 × 2 crossover study was analyzed using a linear mixed model with treatment, period, sequence, and pre-RIPC MI size as fixed effects and subject as a random effect. Healthy volunteers were randomized to the carvedilol or no-carvedilol group. Volunteers were then crossed over to the alternate group after a washout period of six days. Residuals versus fitted values plots were used to check that the error terms (residuals) had a mean of zero and constant variance. The plots did not have any pattern opposed to the equal variance assumption. The normality assumption for the model residuals was checked with histograms and normal quantile–quantile plots of residuals, which seemed to be normally distributed. Analyses were performed using SPSS software (ver. 21.0, IBM, Armonk, NY, USA) and SAS software (ver. 9.2, SAS institute, Cary, NC, USA) for Microsoft Windows. A *p* value < 0.05 was considered statistically significant.

## 5. Conclusions

RIPC-induced cardioprotection was attenuated under sevoflurane or propofol anesthesia in patients undergoing cardiac surgery. There were no differences between sevoflurane and propofol anesthesia. The use of carvedilol also abolished the effects of RIPC in healthy volunteers. These data suggest that sevoflurane, propofol and carvediolol treatment inhibit the effects of RIPC and should be considered in future research. Our research may also explain in part why recent clinical studies of limb RIPC in cardiac surgery have been neutral.

## Figures and Tables

**Figure 1 ijms-20-00269-f001:**
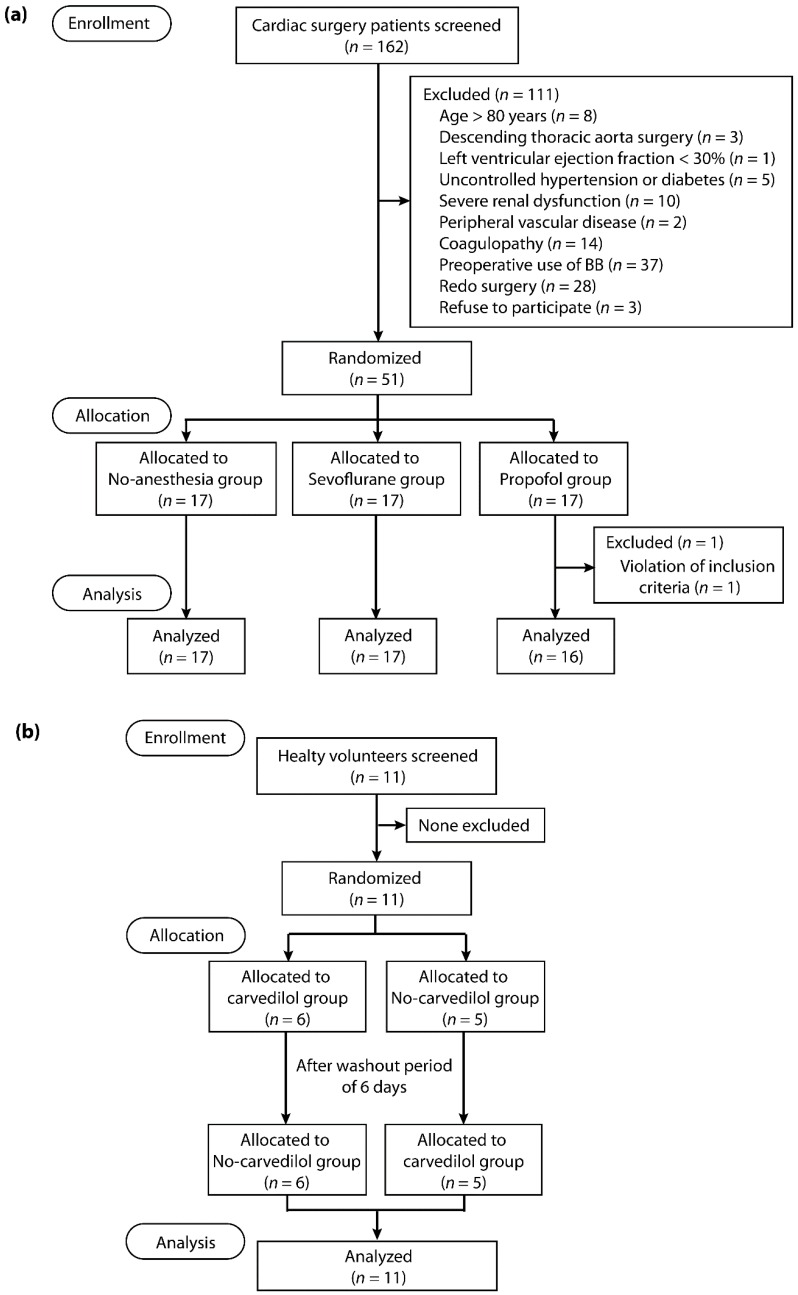
Study flow diagram of: (**a**) cardiac surgery patients; and (**b**) healthy volunteers.

**Figure 2 ijms-20-00269-f002:**
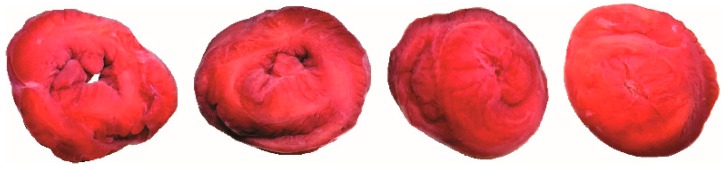
Negative control of staining heart slices after freezing.

**Figure 3 ijms-20-00269-f003:**
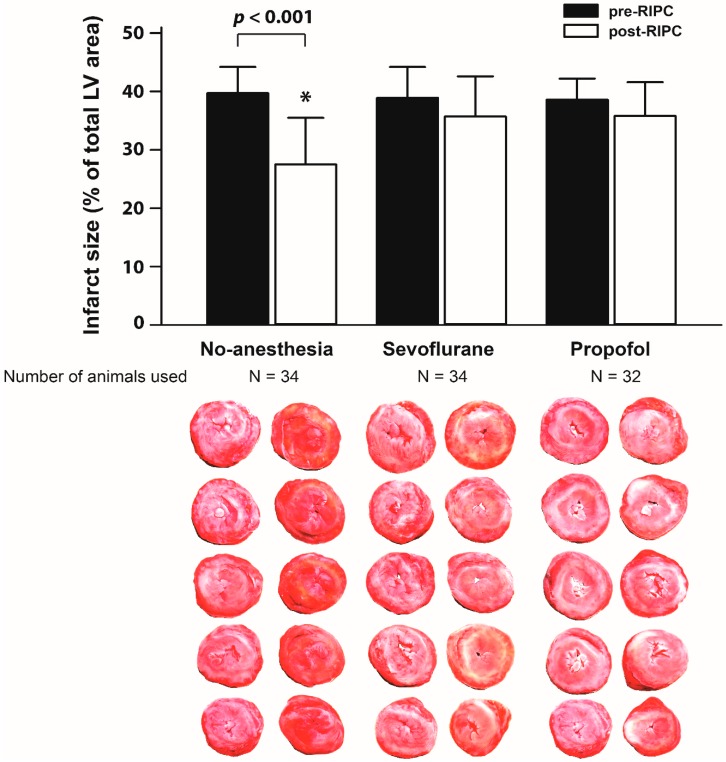
Comparison of pre- and post-RIPC infarct size of the Langendorff isolated rat heart model using dialysate from cardiac surgery patients under different anesthetic implication. Data are presented as mean ± SD. * *p* < 0.05 compared to the pre-RIPC infarct size within a group. LV, left ventricle; RIPC, remote ischemic preconditioning.

**Figure 4 ijms-20-00269-f004:**
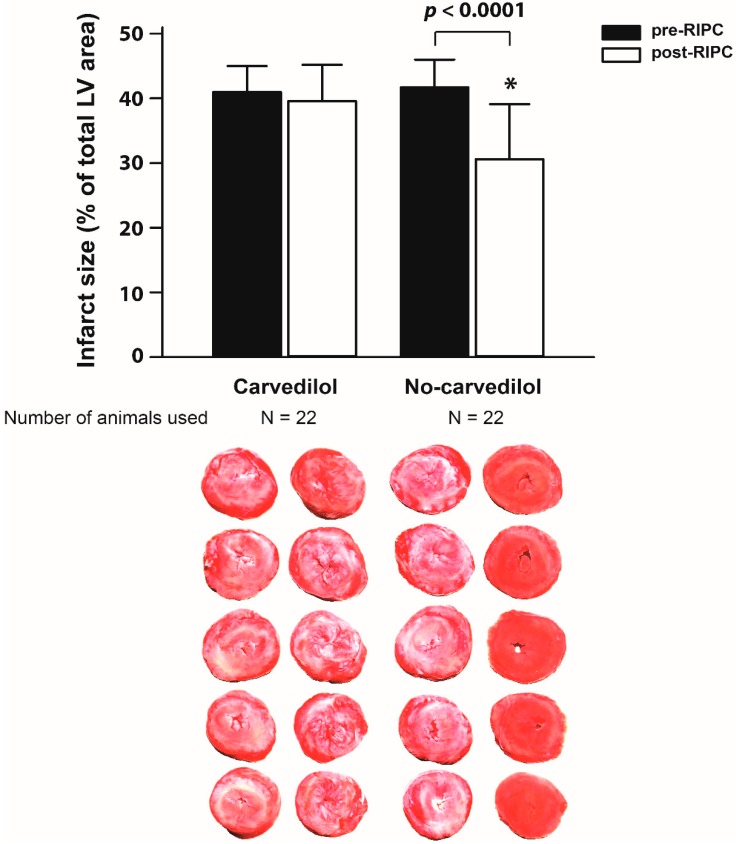
Comparison of pre- and post-RIPC infarct size of the Langendorff isolated rat heart model using dialysate from healthy volunteers with or without taking beta-blocker. Data are presented as mean ± SD. * *p* < 0.05 compared to the pre-RIPC infarct size within a group. LV, left ventricle; RIPC, remote ischemic preconditioning.

**Figure 5 ijms-20-00269-f005:**
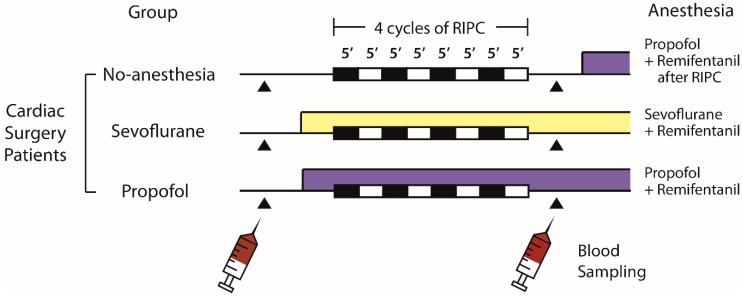
Schematic diagram of the study protocol of cardiac surgery patients. RIPC, remote ischemic preconditioning.

**Figure 6 ijms-20-00269-f006:**
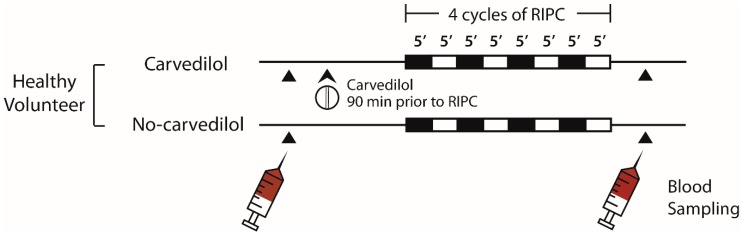
Schematic diagram of the study protocol of healthy volunteers. RIPC, remote ischemic preconditioning.

**Figure 7 ijms-20-00269-f007:**
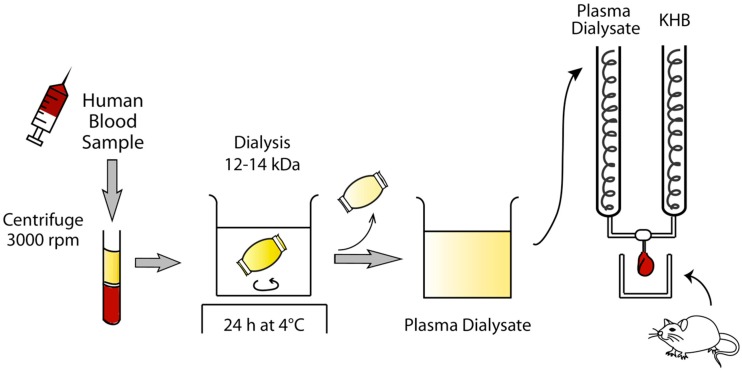
Plasma dialysate preparation and Langendorff rat heart perfusion system. KHB, Krebs–Henseleit buffer.

**Figure 8 ijms-20-00269-f008:**
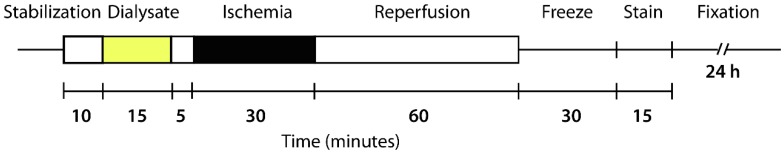
Langendorff rat heart ischemia/reperfusion protocol.

**Table 1 ijms-20-00269-t001:** Characteristics of cardiac surgery patients.

Characteristics	Cardiac Surgical Patients			*p* Value
	No-Anesthesia (*n* = 17)	Sevoflurane (*n* = 17)	Propofol (*n* = 16)	
Age (yr)	66 ± 14	59 ± 15	63 ± 15	0.453
Male sex	6 (35%)	13 (77%)	7 (41%)	0.040
Height (cm)	158.1 ± 11.9	168.7 ± 10.4	160.4 ± 8.3	0.011
Weight (kg)	62.5 ± 10.5	66.4 ± 10.3	62.2 ± 8.1	0.395
BMI (kg/m^2^)	25.0 ± 3.5	23.2 ± 2.4	24.2 ± 3.0	0.242
BSA (m^2^)	1.6 ± 0.2	1.8 ± 0.2	1.7 ± 0.1	0.082
Smoking status (never/current/ex-smoker)	16 (94%)/1 (6%)/0	11 (64%)/3 (18%)/3 (18%)	12 (75%) /2 (12%) /2 (12%)	0.314
Baseline LV EF (%)	62 ± 6	60 ± 7	59 ± 8	0.480
Baseline troponin I (ng/mL)	0.007 ± 0.014	0.018 ± 0.038	0.026 ± 0.049	0.332
Baseline hematocrit (%)	40 ± 3	41 ± 6	39 ± 4	0.357
Baseline eGFR (mL/min/1.73m^2^)	78.1 ± 19.3	86.9 ± 20.6	88.5 ± 27.7	0.371
**Comorbidities**	**No-Anesthesia**	**Sevoflurane**	**Propofol**	
Hypertension	8 (47%)	2 (12%)	6 (37%)	0.075
Diabetes mellitus	2 (12%)	1 (6%)	3 (18%)	0.524
Ischemic heart disease	3 (19%)	1 (6%)	1 (6%)	0.456
Previous PCI	2 (12%)	2 (12%)	0	0.360
Previous stroke	1 (6%)	0	0	0.371
**Preoperative medication**	**No-Anesthesia**	**Sevoflurane**	**Propofol**	
Aspirin	7 (41%)	4 (24%)	2 (13%)	0.165
Clopidogrel	1 (6%)	1 (6%)	0	0.613
ACE inhibitor	0	0	1 (6%)	0.338
ARB	2 (12%)	2 (12%)	3 (18%)	0.802
CCB	9 (53%)	4 (24%)	3 (18%)	0.071
Diuretics	7 (41%)	4 (24%)	6 (37%)	0.520
Nitroglycerin	0	0	1 (6%)	0.338
Digoxin	0	1 (6%)	1 (6%)	0.584
OHA	0	1 (6%)	3 (18%)	0.129
Insulin	0	1 (6%)	0	0.371
Statin	8 (47%)	4 (24%)	5 (31%)	0.337
**Type of surgery**	**No-Anesthesia**	**Sevoflurane**	**Propofol**	0.381
Valve	8 (47%)	10 (59%)	13 (82%)	
Valve + Aorta	7 (41%)	5 (29%)	1 (6%)	
Valve + CABG	1 (6%)	0	1 (6%)	
Aorta	1 (6%)	1 (6%)	1 (6%)	
Other*	0	1 (6%)	0	

Data are presented as mean ± SD or number (%). * Other surgery indicates repair of ventricular septal defect. BMI, body mass index; BSA, body surface area; LV EF, left ventricular ejection fraction; eGFR, estimated glomerular filtration rate; PCI, percutaneous coronary intervention; ACE, angiotensin converting enzyme; ARB, angiotensin receptor blocker; CCB, calcium channel blocker; OHA, oral hypoglycemic agent; CABG, coronary artery bypass graft.

**Table 2 ijms-20-00269-t002:** Characteristics of healthy volunteers.

Characteristics	Healthy Volunteers (*n* = 11)
Age (yr)	27 ± 6
Height (cm)	174.1 ± 4.8
Weight (kg)	71.3 ± 8.2
BMI (kg/m^2^)	23.5 ± 2.2
BSA (m^2^)	1.9 ± 0.1

Data are presented as mean ± SD. BMI, body mass index; BSA, body surface area.

**Table 3 ijms-20-00269-t003:** Postoperative variables in cardiac surgery patients.

Variables	No-Anesthesia (*n* = 17)	Sevoflurane (*n* = 17)	Propofol (*n* = 16)	*p* Value
Peak troponin I in 72 h (ng/mL)	13.07 ± 8.02	11.92 ± 7.33 *	17.44 ± 20.01	0.456
Chest tube drain in 24 h (mL)	697 ± 433	859 ± 538	739 ± 612	0.656
New onset atrial fibrillation	5 (31%)	7 (41%)	7 (47%)	0.671
Intensive care unit stay (h)	42 ± 28	35 ± 27	42 ± 36	0.743
Postoperative hospital stay (days)	12 ± 6	10 ± 4	11 ± 8	0.618

Data are presented as mean ± SD or number (%). * One patient in the sevoflurane group who had intraoperative event related to suspicious left coronary artery injury was excluded from the comparison of postoperative troponin level.

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
