# Peer review of "Sevoflurane, Propofol and Carvedilol Block Myocardial Protection by Limb Remote Ischemic Preconditioning"

_ijms, 2019, doi:10.3390/ijms20020269_

Reviewer 1 Report

The authors report that infarct size reduction allowed by remote post-conditioning was abolished under anesthesia (propofol, sevoflurane) and beta-blockade (carvedilol). The design of the experiment is elegant (remote conditioning induced in volunteers and patients and ischemia-reperfusion secondary performed in isolated rate hearts using dialysate.

1- A control group, i.e., without administration of human dialysate, needs to be added.

2- How do you explain the fact that Pre-RIPC did not show reduction in infarct size? Please discuss.

3- Carvedilol is also known to have proper anti-oxidative properties. Therefore, RPIC blockade might not be due to the antagonism of beta-adrenoceptors. The authors should discuss this aspect. Furthermore, as all investigated substances have particular pharmacological profile, it is hard to say that all anesthetics and beta-blockers block RIPC. Therefore, the title should be modified. Anesthetics and be-blockade should replaced by the Sevoflurane, propofol and carvedilol. Same should be done throughout the text

4- Rat anesthesia was performed using sevoflurane. This might have induced preconditioning and thus could cause interference with the investigated RIPC.

5- Remifentanil was also administered. How can the authors conclude that the observed effect is due to sevoflurane or propofol?

6- Heart were frozen before cutting and TTC incubation. Frozen itself might induced cell death. The authors should provide evidence that this procedure itself have not induced TTC negative staining.

7- Did calculation of infarct size take into account the weight of each slice ?

Author Response

Point 1: A control group, i.e., without administration of human dialysate, needs to be added.

Response 1: Thank you for the comments. We have preliminary data on MI size of rat hearts undergoing ischemia-reperfusion injury without using human dialysate. Three rat hearts were used and perfused with Krebs-Henseleit buffer solution for 30 min before 30-min no-flow global ischemia and subsequent 60-min reperfusion on a Langendorff apparatus. We presented this data as a control group without administration of human dialysate in the revised manuscript as follows (page 7, lines 143–145):

According to our preliminary data, MI size of rat hearts (n=3) perfused for 30 min and then underwent IR injury, consisting of 30-min global ischemia followed by 60-min reperfusion, without treatment of human dialysate was 41.1±6.8%.

Point 2:How do you explain the fact that Pre-RIPC did not show reduction in infarct size? Please discuss.

Response 2:Thank you for the comments. Pre-RIPC dialysate is considered not to contain any protective mediators because it was obtained from the subjects before application of RIPC. Therefore, it is not expected that treatment of pre-RIPC dialysate would elicit any reduction in infarct size at least by the influence of RIPC. Accordingly, there were no differences of MI sizes treated with pre-RIPC dialysate between patient groups and healthy volunteer groups.

Point 3:Carvedilol is also known to have proper anti-oxidative properties. Therefore, RPIC blockade might not be due to the antagonism of beta-adrenoceptors. The authors should discuss this aspect. Furthermore, as all investigated substances have particular pharmacological profile, it is hard to say that all anesthetics and beta-blockers block RIPC. Therefore, the title should be modified. Anesthetics and be-blockade should replaced by the Sevoflurane, propofol and carvedilol. Same should be done throughout the text.

Response 3:Thank you for the comments. We agree with your opinion. As we mentioned in our Discussion section (page 11, lines 267–268), carvedilol, like propofol, acts as a free radical scavenger and has an anti-oxidative property. Thereby we cannot completely exclude a possibility that anti-oxidative properties of carvedilol may have a role in inhibiting RIPC effects in this study. Thus we added description on this issue in the Discussion as follows (page 11, lines 268–270):

Thus, there is also a possibility that carvedilol might have blocked effects of RIPC by its anti-oxidative properties regardless of blockade of beta-adrenoceptors in this study.

We also modified the title. “Anesthetics and beta-blockers” were replaced by “Sevoflurane, propofol and carvedilol” in the title, and also throughout the text, according to the reviewer’s comments.

Point 4:Rat anesthesia was performed using sevoflurane. This might have induced preconditioning and thus could cause interference with the investigated RIPC.

Response 4:Thank you for the comments. As we described in the Methods section (page 14, lines 429–430), the duration of sevoflurane administration to induce anesthesia in rats was less than two minutes before the rat heart was immediately excised and mounted on the Langendorff apparatus. Although we cannot exclude the possibility of any effects of inhaled sevoflurane completely, any additive cardioprotective effects induced by a brief period of inhalation might not have affected the results on MI size differently among the treatment groups, as 90% of rats were treated equally with the same anesthetic in the cardiac surgery study, and 100% of rats were anesthetized with the same drug in the healthy volunteer study. We have mentioned this issue in the revised manuscript as follows (page 14, lines 430–433): 

Therefore, although we cannot exclude the possibility of any effects of inhaled sevoflurane in rats completely, any additive cardioprotective effects induced by this brief period of inhalation would have affected MI size equally among the treatment groups.

Point 5:Remifentanil was also administered. How can the authors conclude that the observed effect is due to sevoflurane or propofol?

Response 5:Thank you for the comments. In accordance with the reviewer’s comment, remifentanil is known to have cardioprotective effects and may affect preconditioning-induced infarct size changes following various conditions. We added description regarding remifentanil used in the study protocol in the Discussion section of the revised manuscript, as follows (page 11, lines 277–287):

In addition, opioid receptors are implicated in myocardial conditioning and cardioprotective responses [46]. Continuous administration of remifentanil provided cardioprotection in in vitro human myocardium against hypoxia reoxygenation stress [47]. On the other hand, sustained administration of remifentanil inhibited preconditioning-induced infarct limitation following one cycle of preconditioning (5-min ischemia and 5-min reperfusion), but did not affect MI size reduction elicited by two cycles of preconditioning in rabbit hearts [48]. In this study, we investigated the effects of sevoflurane and propofol on RIPC-induced cardioprotection in a pragmatic manner that is commonly used in clinical anesthetic practice for cardiac surgery patients. Nevertheless, we cannot completely exclude any influence of remifentanil administered during RIPC in this study, and further studies are required to determine the effects of administered remifentanil during RIPC in cardiac surgery patients.

Point 6:Heart were frozen before cutting and TTC incubation. Frozen itself might induced cell death. The authors should provide evidence that this procedure itself have not induced TTC negative staining.

Response 6:Thank you for the comments. Freezing heart slices before cutting is well-established process in the ischemia-reperfusion model that is commonly used with the Langendorff perfusion system. We have added appropriate references regarding the freezing step to the revised manuscript (page 15, line 450).

Added references are:

[ref #50] Bell R.M.; Mocanu M.M.; Yellon D.M. Retrograde heart perfusion: the Langendorff technique of isolated heart perfusion. J. Mol. Cell. Cardiol. 2011, 50, 940–50. https://doi.org/10.1016/j.yjmcc.2011.02.018.

[ref #51] Heinen A.; Behmenburg F.; Aytulun A.; Dierkes M.; Zerbin L.; Kaisers W.; Schaefer M.; Meyer-Treschan T.; Feit S.; Bauer I.; et al. The release of cardioprotective humoral factors after remote ischemic preconditioning in humans is age- and sex-dependent. J. Transl. Med. 2018, 16, 112. https://doi.org/10.1186/s12967-018-1480-0.

Point 7:Did calculation of infarct size take into account the weight of each slice ?

Response 7:The infarct size was expressed as a percentage of the total area of the LV without weighing the slices. We have modified the description on calculation of MI size in the revised manuscript to clarify this and have added appropriate references from the literature (page 15, lines 457).

The added references are: 

[ref #20] McDonald M.A.; Braga J.R.; Li J.; Manlhiot C.; Ross H.J.; Redington A.N. A randomized pilot trial of remote ischemic preconditioning in heart failure with reduced ejection fraction. PLoS One 2014, 9, e105361. https://doi.org/10.1371/journal.pone.0105361.

[ref #51] Heinen A.; Behmenburg F.; Aytulun A.; Dierkes M.; Zerbin L.; Kaisers W.; Schaefer M.; Meyer-Treschan T.; Feit S.; Bauer I.; et al. The release of cardioprotective humoral factors after remote ischemic preconditioning in humans is age- and sex-dependent. J. Transl. Med. 2018, 16, 112. https://doi.org/10.1186/s12967-018-1480-0.

Reviewer 2 Report

The MS of Y.J. Cho et al., investigates the role of anesthetics and beta-blockers on myocardial protection by limb remote ischemic preconditioning. Dialyzed plasma samples from cardiac surgery patients were used in a xenograft way in an isolated rat heart model of acute myocardial ischemia/reperfusion injury. The paper basically investigates actual and clinically current issues of the field, however, there are several concerns about the design of the study and its message is also not clear. From an ethical point of view the study raises also severe concerns.

Major comments:

1. The main concern of the paper is that the authors claim that there was a group of patients underwent cardiac surgery without anesthesia. It is not clear that the “awake state” was only before the cardiac surgery or during the whole procedure. What kind of pain killers were given to the patients?

2. Xenograft experiments between species is an interesting field, however the authors choice of species are questionable. Why did the authors choose human-rat combinations? Interspecies differences between humans and rodents are too large to be a good comparative model. Indeed, isolated rodent hearts are easy to handle on a Langendorff perfusion system and do not require much plasma dialysate. However, giving them human plasma from cardiac surgery patients without anesthesia, seems to be a weird approach from both translational and ethical aspects as well. Why the authors did not choose pig or dog (or other big animal) model as donors instead of humans?

3. The authors state that their only inclusion criteria were the age between 20 and 80 and undergoing cardiac surgery using cardiopulmonary bypass. It is not clear what was the reasonable explanation to choose the above inclusion criteria. If the age range was so wide already why they exclude patients over 80? Furthermore, from 162 enrolled patients 111 (more than 2/3!) were excluded. Why did the authors not enroll patients using stricter primary criteria?

4. It is also not clear, why the authors have enrolled such a diverse study population including valvular disease patients, patients with coronary stenosis, and patients with aortic defects? What does “others” mean in Table 1 last row? Please, clarify.

5. Why did the authors use no-flow ischemia instead of coronary occlusion? The translatability of global ischemia to human circumstances is inferior compared to coronary occlusion. Authors used the conventional TTC staining for showing infarct size, however, they used only 60 min reperfusion. It is well-known that the reliability of TTC staining depends on duration of reperfusion, the longer reperfusion is applied the more reliable infarct size can be shown. Therefore, most papers apply at least 120 min reperfusion. What was the reason to reduce it?

6. In figure 1, part b the allocation of volunteers, the 2x2 crossover study design is not clear. What do the crossing arrows mean? Even the description in section 4.9 - statistical analysis – does not help to understand it. Please clarify and rewrite this section accordingly.

7. Table1 shows inconsistent data, such as a) Baseline data are missing in case of healthy volunteers (eg smoking status, hematocrit, etc) b) p values are not shown, therefore, the reader cannot see the statistical comparison of the experimental groups c) In case of healthy patients, the volunteers were males only. However, in the first study (cardiac surgery) 24 out of 50 patients were females. What is the reason for choosing only male individuals for enrollment?

8. In Figures 2 and 3 the standard deviation (I suppose it is SD according to the 4.9 section, but it is not written in the figure legends) is too high, I do not see how it could be significantly different. The number of animals used in the different groups are not provided. The representative images of the rat heart slices are of poor quality; on the given photos the brick red color and pale white color is really hard to be distinguished. 9. In section 4.7: authors used different types of anesthesia. Beyond the comparability issues of the two different way of anesthesia, the main concern is that the authors investigated the role of sevoflurane in interfering with the cardioprotection of RIPC in humans and probably even in hearts treated with plasma dialysates from control (not anesthetized) patients, they administered exactly the same drug to induce anesthesia in rats. However, the latter is not clearly declared, if the hearts were treated with a mixture of human plasma dialysates or each heart was treated only with one certain plasma dialysate derived from one patient. This should also be described clearly.

10. In case of animal model the exclusion criteria were time, contractile function, arrhythmia and heart rate. How many animals were excluded?

Minor comment:

• The statement of “The protective effects of RIPC can be transferred between species”, is supported by only one reference. Please, add further relating references to this section, if applicable.

• Table1: in propofol group the number of involved patients is n=16 in the title line, however, at the last section of this table, at the type of surgery, the number of patients is 17 in this group (14+1+1+1). Which is the real data?

Author Response

Major comments:

Point 1:The main concern of the paper is that the authors claim that there was a group of patients underwent cardiac surgery without anesthesia. It is not clear that the “awake state” was only before the cardiac surgery or during the whole procedure. What kind of pain killers were given to the patients?

Response 1:Thank you for the comments. The “awake state” indicates that only the RIPC procedure was performed in the awake state- following this, cardiac surgery proceed under anesthesia as per normal clinical practice. After RIPC in the control group, general anesthesia was provided to the patients during the surgery, and this is also stated in the manuscript in the Methods section, 4.4 Cardiac surgery study design, as follows (page 12, lines 362–363):

In the no-anesthesia group, RIPC was performed in the awake state before anesthesia on the day of surgery (Figure 4).

We added additional details regarding the anesthetic protocol in the no-anesthesia group to clarify that anesthesia was provided after RIPC for cardiac surgery in the no-anesthesia group (page 12, lines 363–364) as follows:

General anesthesia was provided to the patients in the no-anesthesia group after RIPC study protocol during the surgery.

Also, we modified Figure 4 to demonstrate anesthesia provided in the no-anesthesia group after RIPC (page 13, Figure 4). 

There was no group of patients undergoing cardiac surgery without anesthesia.

Point 2:Xenograft experiments between species is an interesting field, however the authors choice of species are questionable. Why did the authors choose human-rat combinations? Interspecies differences between humans and rodents are too large to be a good comparative model. Indeed, isolated rodent hearts are easy to handle on a Langendorff perfusion system and do not require much plasma dialysate. However, giving them human plasma from cardiac surgery patients without anesthesia, seems to be a weird approach from both translational and ethical aspects as well. Why the authors did not choose pig or dog (or other big animal) model as donors instead of humans?

Response 2:Thank you for the comments. The primary research question was whether several promising confounding factors interfere with RIPC in patients undergoing cardiac surgery because there has been mixed results on RIPC in patients undergoing cardiac surgery, unlike patients undergoing primary coronary intervention who do not require general anesthesia. Accordingly, the primary purpose of the current study was to investigate whether anesthetics (representatively, sevoflurane and propofol) interfere with RIPC in patients undergoing cardiac surgery, not in other animals. Therefore we obtained plasma dialysate from patients receiving RIPC before scheduled cardiac surgery and then examined the effects of RIPC by comparing infarct size of rat hearts in Langendorff ischemia-reperfusion injury model.

Point 3:The authors state that their only inclusion criteria were the age between 20 and 80 and undergoing cardiac surgery using cardiopulmonary bypass. It is not clear what was the reasonable explanation to choose the above inclusion criteria. If the age range was so wide already why they exclude patients over 80? Furthermore, from 162 enrolled patients 111 (more than 2/3!) were excluded. Why did the authors not enroll patients using stricter primary criteria?

Response 3:Thank you for the comments. We planned to recruit study population of adult cardiac surgery patients and we excluded over-octogenarians, as any changes in neuronal or humoral responses to RIPC in these late elderly patients were not identified. However, we did not restrict the age limitation under 65 or younger because most patients receiving cardiac surgery are still old-aged. The mean age of the overall included patients in this study was 63 years. 

Moreover, most cardiac surgery includes the use of cardiopulmonary bypass strategy and surgeries without using cardiopulmonary bypass might have heterogeneous disease entity, thereby increasing heterogeneity of study population.

We added more explanation on inclusion criteria in the revised manuscript as follows (page 12, lines 328–329):

We did not include over-octogenarian patients, as any possible changes in neuronal or humoral responses to RIPC in these patients were not clearly identified.

Point 4:It is also not clear, why the authors have enrolled such a diverse study population including valvular disease patients, patients with coronary stenosis, and patients with aortic defects? What does “others” mean in Table 1 last row? Please, clarify.

Response 4:Thank you for the comments. We intended to include all kinds of adult cardiac surgery using cardiopulmonary bypass in this study and did not focus on the specific type of surgery. Moreover, the main study intervention (RIPC) is completed before the main surgical procedures were started. Therefore, the primary study outcome (changes in infarct size) is determined regardless of what kind of surgery the patients underwent.

“Other” surgery included repair of ventricular septal defect in one patient in the sevoflurane group. Thus we separated the mixed expression of “aorta + others” into “aorta” and “other” surgery, and accordingly, we modified Table 1 (revised manuscript, pages 5-6). Also, we clarified that “other” surgery indicates repair of ventricular septal defect in the table legends, according to the reviewer’s comments. 

Point 5:Why did the authors use no-flow ischemia instead of coronary occlusion? The translatability of global ischemia to human circumstances is inferior compared to coronary occlusion. Authors used the conventional TTC staining for showing infarct size, however, they used only 60 min reperfusion. It is well-known that the reliability of TTC staining depends on duration of reperfusion, the longer reperfusion is applied the more reliable infarct size can be shown. Therefore, most papers apply at least 120 min reperfusion. What was the reason to reduce it?

Response 5:Thank you for the comments. 

We agree with the reviewer’s opinion that the local ischemia is more conventional procedure in rat hearts ischemic models. However, there also are many publications in which global ischemia was induced in rat hearts to successfully show infarct size determination under TTC staining after ischemia-reperfusion injury (Ferrera et al, 2009; Bell et al, 2011; Heinen et al, 2018).

More importantly, unlike STEMI patients undergoing coronary intervention in which local ischemic insult is more apparent, cardiac surgery patients undergoing cardiopulmonary bypass are more relevant to global ischemia than occlusion of one coronary artery. 

In regard of period of reperfusion, there has been much recent evidence that 60-min reperfusion to assess infarct size under TTC staining in rat hearts are reliable and there were no differences in infarct area between 60-min and 120-min period of reperfusion in rat hearts (Ferrera et al, 2009; Bell et al, 2011; Heinen et al, 2018).

We added appropriate references in the revised manuscript (page 14, line 441).

Added references are:

[ref #49] Ferrera R.; Benhabbouche S.; Bopassa J.C.; Li B.; Ovize M. One hour reperfusion is enough to assess function and infarct size with TTC staining in Langendorff rat model. Cardiovasc. Drugs Ther. 2009, 23, 327–331. https://doi.org/10.1007/s10557-009-6176-5.

[ref #50] Bell R.M.; Mocanu M.M.; Yellon D.M. Retrograde heart perfusion: the Langendorff technique of isolated heart perfusion. J. Mol. Cell. Cardiol. 2011, 50, 940–950. https://doi.org/10.1016/j.yjmcc.2011.02.018.

[ref #51] Heinen A.; Behmenburg F.; Aytulun A.; Dierkes M.; Zerbin L.; Kaisers W.; Schaefer M.; Meyer-Treschan T.; Feit S.; Bauer I.; et al. The release of cardioprotective humoral factors after remote ischemic preconditioning in humans is age- and sex-dependent. J. Transl. Med. 2018, 16, 112. https://doi.org/10.1186/s12967-018-1480-0.

Point 6:In figure 1, part b the allocation of volunteers, the 2x2 crossover study design is not clear. What do the crossing arrows mean? Even the description in section 4.9 - statistical analysis – does not help to understand it. Please clarify and rewrite this section accordingly.

Response 6:Thank you for the comments. We modified Figure 1 part b (page 3) to clarify the crossover study design more clearly and rewrote the section 4.9. Statistical analysis to help to understand the study design more easily, according to the reviewer’s comment, as follows (page 16, lines 505–507):

Healthy volunteers were randomized to the carvedilol or no-carvedilol group. Volunteers were then crossed over to the alternate group after a washout period of 6 days.

Point 7:Table1 shows inconsistent data, such as a)Baseline data are missing in case of healthy volunteers (eg smoking status, hematocrit, etc) b)        p values are not shown, therefore, the reader cannot see the statistical comparison of the experimental groups c)     In case of healthy patients, the volunteers were males only. However, in the first study (cardiac surgery) 24 out of 50 patients were females. What is the reason for choosing only male individuals for enrollment?

Response 7:Thank you for the comments. Originally, the two studies in cardiac surgery patients and healthy volunteers were separate studies with isolated study design each approved by IRB.

a) Some of the data in healthy volunteers are not missing, but were not assessed according to the study protocol approved by IRB. We separated the table regarding participants’ characteristics into two tables (revised Tables 1 and 2, pages 5-6) for cardiac surgery patients and healthy volunteers, respectively, to minimize unnecessary comparison between the two separate study groups.

b) We added p values in the revised Table 1 for comparison among cardiac surgery patient groups.

c) The purpose of the healthy volunteer study was to evaluate the effects of carvedilol on changes in infarct size after RIPC, so we avoided other possible confounders that might affect our results. According to evidence from literatures, basic physiological differences, such as difference in hormones and drug-metabolizing enzymes or changes in total body water, in males and females can make response to cardiovascular drugs different (Abbas et al., 2014; Baris et al., 2006). In a previous study by Abbas et al., after single oral dose of 12.5 mg of carvedilol, pharmacokinetic parameters and bioavailability were different between healthy male and female volunteers.

References are: 

Abbas M.; Khan A.M.; Riffat S.; Tipu M.Y.; Nawaz H.A.; Usman M. Assessment of sex differences in Pharmacokinetics of carvedilol in human. Pak. J. Pharm. Sci. 2014, 27, 1265–9.

Baris N.; Kalkan S.; Guneri S.; Bozdemir V.; Guven H. Influence of carvedilol on serum digoxin levels in heart failure: is there any gender difference? Eur. J. Clin. Pharmacol. 2006, 62, 535–8.

Point 8:In Figures 2 and 3 the standard deviation (I suppose it is SD according to the 4.9 section, but it is not written in the figure legends) is too high, I do not see how it could be significantly different. The number of animals used in the different groups are not provided. The representative images of the rat heart slices are of poor quality; on the given photos the brick red color and pale white color is really hard to be distinguished.

Response 8:Thank you for the constructive comments. We indicated that the data are presented as means ± SD in the figure legends of Figures 2 and 3. We modified the axes in the figures to be started from zero so that the readers could see more clearly not only the mean and the SD of each data, but also the differences among the groups. And we also checked the comparisons of IS differences among the groups are significantly different according to the appropriate statistics. We added more detailed descriptions regarding the differences between pre- and post-RIPC MI sizes with presenting the mean differences and 95% CIs in the revised manuscript as follows (page 7, lines 148–149; page 8, lines 162–166):

However, MI size was significantly reduced after RIPC in the no-anesthesia group (post-RIPC MI size, 27.5±8.0%; mean difference -12.1%; 95% CI -17.4%, -6.9%; p<0.001),< span="">

In the healthy volunteer study, there was a significant reduction in MI size with RIPC in the no-carvedilol group (41.7±4.3% pre-RIPC vs. 30.6±8.5% post-RIPC; mean difference -10.6%; 95% CI -14.7%, -6.5%; p<0.0001) (Figure 3). However, in the carvedilol group, there was no significant difference in MI size (41.0±4.0% pre-RIPC vs. 39.6±5.6% post-RIPC; mean difference -1.6%; 95% CI -5.9%, 2.8%; p=0.452).

We provided the number of animals used in Figures 2 and 3 (pages 7 and 8), as well as in the Methods section, 4.7 Langendorff-perfused rat heart model, in the revised manuscript as follows (page 14, lines 426–429): 

Among the 100 anesthetized rats used in the cardiac surgery study, 10 were given pentobarbital sodium and the remaining rats were given sevoflurane due to the shortage of pentobarbital in our animal laboratory. Forty-four rats used in the healthy volunteer study were anesthetized with sevoflurane.

We provided newly selected photos of the rat heart slices that are more clearly distinguishable between the red and the white color in Figures 2 and 3 (revised manuscript, pages 7 and 8), according to the reviewer’s comment.

Point 9:In section 4.7: authors used different types of anesthesia. Beyond the comparability issues of the two different way of anesthesia, the main concern is that the authors investigated the role of sevoflurane in interfering with the cardioprotection of RIPC in humans and probably even in hearts treated with plasma dialysates from control (not anesthetized) patients, they administered exactly the same drug to induce anesthesia in rats. However, the latter is not clearly declared, if the hearts were treated with a mixture of human plasma dialysates or each heart was treated only with one certain plasma dialysate derived from one patient. This should also be described clearly.

Response 9:Thank you for the comments. 

As mentioned in the Methods section (page 14, lines 429–430), the duration of sevoflurane administration to induce anesthesia in rats was less than two minutes before the rat heart was immediately excised and mounted on the Langendorff apparatus. Although we cannot exclude the possibility of any effects of inhaled sevoflurane completely, any additive cardioprotective effects induced by a brief period of inhalation might not have affected the results on MI size differently among the treatment groups, as 90% of rats were treated evenly with the same anesthetic in the cardiac surgery study, and 100% of rats were anesthetized with the same drug in the healthy volunteer study. We also mentioned on this issue in the revised manuscript as follows (page 14, lines 430–433): 

Therefore, although we cannot exclude the possibility of any effects of inhaled sevoflurane in rats completely, any additive cardioprotective effects induced by a brief period of inhalation might not have affected the results on MI size differently among the treatment groups.

We treated each heart with one certain plasma dialysate derived from one patient so that the effects of RIPC performed in each patient would be evaluated from each experiment. We added more details on this issue in the revised manuscript to make the methods more clearly according to the reviewer’s recommendation, as follows (page 14, lines 441–442):

Each heart was treated only with one certain plasma dialysate derived from one human subject.

Point 10:In case of animal model the exclusion criteria were time, contractile function, arrhythmia and heart rate. How many animals were excluded?

Response 10:We excluded two experiments from the analysis: one was due to sustained poor ventricular contractile function and the other was due to poor contractile function combined with sustained ventricular fibrillation. We also added the details on excluded experiments in the revised manuscript as follows (page 4, lines 100–102):

There were an experimental failure with the Langendorff model each in the no-anesthesia and carvedilol group, due to the sustained poor ventricular contractile function and the poor contractile function combined with sustained ventricular fibrillation, respectively.

Minor comments:

Point 1:The statement of “The protective effects of RIPC can be transferred between species”, is supported by only one reference. Please, add further relating references to this section, if applicable.

Response 1:Thank you for the comments. We added further relating references, in which cardioprotective effects of RIPC, transferred across species with human plasma-derived dialysate were evaluated, in the revised manuscript, page 2, line 78.

Added references are: 

[ref #18] Jensen, R.V.; Stottrup, N.B.; Kristiansen, S.B.; Botker, H.E. Release of a humoral circulating cardioprotective factor by remote ischemic preconditioning is dependent on preserved neural pathways in diabetic patients. Basic Res. Cardiol. 2012, 107, 285. https://doi.org/10.1007/s00395-012-0285-1.

[ref #19] Heusch, G.; Botker, H.E.; Przyklenk, K.; Redington, A.; Yellon, D. Remote ischemic conditioning. J. Am. Coll. Cardiol. 2015, 65, 177–195. https://doi.org/10.1016/j.jacc.2014.10.031.

[ref #20] McDonald M.A.; Braga J.R.; Li J.; Manlhiot C.; Ross H.J.; Redington A.N. A randomized pilot trial of remote ischemic preconditioning in heart failure with reduced ejection fraction. PLoS One 2014, 9, e105361. https://doi.org/10.1371/journal.pone.0105361.

Point 2:Table1: in propofol group the number of involved patients is n=16 in the title line, however, at the last section of this table, at the type of surgery, the number of patients is 17 in this group (14+1+1+1). Which is the real data?

Response 2:We double-checked the data included in the analyses. And we corrected the number of patients who underwent valve surgery to be 13 in the propofol group. (pages 5-6, Table 1) After thorough checking other data, we corrected number of never smoker to be 12 (75%), proportion of patients with hypertension and taking diuretics to be 37%, and taking statin to be 31%, in the propofol group (Table 1).

Round  2

Reviewer 1 Report

Thank you for your revised version and the answer to the questions. However :

- as previously mentioned, calculation of infarct size needs to take into account the weight of the heart.

- as previously mentioned, please provide evidences that the freezing process does not produce TTC negative staining.

Academic Editor Notes

We should ask the authors to give some more information for control group, use of other interfering substances, and some methodological clarification according to reviewer 1.

Response: Thank you for the comments. We did not use any other interfering substances for the control group, thus we have described more details in the revised manuscript as follows (page 7, lines 96–99):

According to our preliminary data, MI size of rat hearts (control group, n=3) perfused for 30 min and then underwent IR injury, consisting of 30-min global ischemia followed by 60-min reperfusion, without treatment of human dialysate was 41.1±6.8%. For the control group, no other interfering substances but only neutral buffer solution was used to perfuse the isolated rat heart.

Reviewer 1 (Round 2)

Point 1: as previously mentioned, calculation of infarct size needs to take into account the weight of the heart.

Response 1: Thank you for the comment. The infarct size of the animal hearts in this study was expressed as a percentage of the total area of the LV, as many previous studies on myocardial IR injury did (McDonald et al, 2014; Heinen et al, 2018; Redington et al, 2013; Lateef et al, 2015). We did not weigh the heart, but we cut the heart into 5–6 slices with the same thickness using 1 mm Rat Heart Matrix (CellPoint Scientific, Gaithersburg, MD, USA). The MI size of the individual heart was calculated as the mean of the infarct sizes of all the slices from each heart. Similarly, previous investigators, including McDonald et al (2014), Heinen et al (2018), and Redington et al (2013), expressed the MI size of the experimental animal hearts subjected to IR injury as percentages of the total area at risk using planimetric determination. 

We have added further references from the literature in the revised manuscript (page 14, line 344).

References are: 

[ref #20] McDonald M.A.; Braga J.R.; Li J.; Manlhiot C.; Ross H.J.; Redington A.N. A randomized pilot trial of remote ischemic preconditioning in heart failure with reduced ejection fraction. PLoS One2014,9, e105361. https://doi.org/10.1371/journal.pone.0105361.

[ref #51] Heinen A.; Behmenburg F.; Aytulun A.; Dierkes M.; Zerbin L.; Kaisers W.; Schaefer M.; Meyer-Treschan T.; Feit S.; Bauer I.; et al. The release of cardioprotective humoral factors after remote ischemic preconditioning in humans is age- and sex-dependent. J. Transl. Med2018, 16, 112. https://doi.org/10.1186/s12967-018-1480-0.

[ref #52] Lateef R.; Al-Masri A.; Alyahya A. Langendorff’s isolated perfused rat heart technique: a review. Int. J. Basic Clin. Pharmacol.2015,4, 1314–22. https://doi.org/10.18203/2319-2003.ijbcp20151381.

[ref #53] Redington K.L.; Disenhouse T.; Li J.; Wei C.; Dai X.; Gladstone R.; Manlhiot C.; Redington A.N. Electroacupuncture reduces myocardial infarct size and improves post-ischemic recovery by invoking release of humoral, dialyzable, cardioprotective factors. J. Physiol. Sci2013,63, 219–23. https://doi.org/10.1007/s12576-013-0259-6.

Point 2:as previously mentioned, please provide evidences that the freezing process does not produce TTC negative staining.

Response 2:Thank you for the comment. Freezing heart slices before cutting is a well-established process in the ischemia-reperfusion model using the Langendorff heart perfusion system, as in many other published research papers we provided with references (Bell et al, 2011; Heinen et al, 2018; Lateef et al, 2015). Several investigators (McDonald et al, 2014; Redington et al, 2013) even froze the hearts at –80°C, much lower temperature than ours (–20°C), before they cut the experimental hearts and stained with TTC, eliciting significant differences (that means TTC positive results) in comparing infarct sizes.

Consistently, we observed obvious differences between brick red and pale white colors from our stained heart slices, as we provided the photos of the experimental heart slices in Figures 2 and 3 (manuscript pages 7–8). If the heart frozen and eventually dead, we could have not obtained any red colors from our heart slices after TTC staining. However we successfully calculated infarct sizes by distinguishing pale white colors from red colors, and compared the MI size among the treatment groups.

We also have added relevant references regarding the freezing step to the revised manuscript (page 14, line 337).

References are:

[ref #50] Bell R.M.; Mocanu M.M.; Yellon D.M. Retrograde heart perfusion: the Langendorff technique of isolated heart perfusion. J. Mol. Cell. Cardiol.2011,50, 940–50. https://doi.org/10.1016/j.yjmcc.2011.02.018.

[ref #51] Heinen A.; Behmenburg F.; Aytulun A.; Dierkes M.; Zerbin L.; Kaisers W.; Schaefer M.; Meyer-Treschan T.; Feit S.; Bauer I.; et al. The release of cardioprotective humoral factors after remote ischemic preconditioning in humans is age- and sex-dependent. J. Transl. Med.2018,16, 112. https://doi.org/10.1186/s12967-018-1480-0.

[ref #52] Lateef R.; Al-Masri A.; Alyahya A. Langendorff’s isolated perfused rat heart technique: a review. Int. J. Basic Clin. Pharmacol.2015,4, 1314–22. https://doi.org/10.18203/2319-2003.ijbcp20151381.

[ref #20] McDonald M.A.; Braga J.R.; Li J.; Manlhiot C.; Ross H.J.; Redington A.N. A randomized pilot trial of remote ischemic preconditioning in heart failure with reduced ejection fraction. PLoS One2014,9, e105361. https://doi.org/10.1371/journal.pone.0105361.

[ref #53] Redington K.L.; Disenhouse T.; Li J.; Wei C.; Dai X.; Gladstone R.; Manlhiot C.; Redington A.N. Electroacupuncture reduces myocardial infarct size and improves post-ischemic recovery by invoking release of humoral, dialyzable, cardioprotective factors. J. Physiol. Sci2013,63, 219–23. https://doi.org/10.1007/s12576-013-0259-6.

Reviewer 2 Report

The revised manuscript can be accepted for publication

Author Response

Thank you for your decision.

Round  3

Reviewer 1 Report

Thanks for author's reply.

1- Regarding the calculation of infarct size, I still disagree with the authors. Weight of the slices needs to be taken into account. Nevertheless as it is indicated that infarct size was calculated as percentage of LV area, I'm only asking that the authors add in their discussion as a limitation of the sudy that their infarct size calculation did not take into account the weight of the slices and that only areas were used.

2- Regarding the freezing process before cutting the heart into slices, the authors did not properly answered the comment. The authors MUST provide representative pictures showing that freezing the heart does not induces negative TTC staining in SHAM hearts (just excise the heart, freeze, cut and TTC). Two hearts are enough. It is very simple and very quick experiments.

Author Response

Point 1: Regarding the calculation of infarct size, I still disagree with the authors. Weight of the slices needs to be taken into account. Nevertheless as it is indicated that infarct size was calculated as percentage of LV area, I'm only asking that the authors add in their discussion as a limitation of the study that their infarct size calculation did not take into account the weight of the slices and that only areas were used.

Response 1: We totally agree with the comment. According to the reviewer’s comments, we have added limitation of the study in the Discussion section regarding the calculation of the infarct size, in the revised manuscript as follows (page 11, lines 227–228):

Fourth, calculation of the MI size of rat hearts did not take into account the weight of the heart slices, and only areas were used.

Point 2:Regarding the freezing process before cutting the heart into slices, the authors did not properly answered the comment. The authors MUST provide representative pictures showing that freezing the heart does not induces negative TTC staining in SHAM hearts (just excise the heart, freeze, cut and TTC). Two hearts are enough. It is very simple and very quick experiments.

Response 2:Thank you for the constructive comment. We totally agree with the opinion. According to the reviewer’s recommendation, we performed experiments for negative control immediately after we got the reviewer’s comments. And we have added description on the negative control of TTC staining in sham hearts using two rats in the revised manuscript as follows (page 7, lines 100–105): 

Also, we confirmed that the freezing process before cutting and staining the heart slices did not induce negative staining in sham hearts (n=2) by performing negative control experiments – the hearts were excised, frozen at -20ºC for 30 min, cut into slices and then stained (Figure 2).